Soil properties and microbial communities of spring maize filed in response to tillage with straw incorporation and nitrogen fertilization in northeast China

Sui Pengxiang 1 2 3
Tian Ping 1
Wang Zhengyu 1
Lian Hongli 1
Yang Yadong 4
Ma Ziqi 1
Jiang Ying 1 jiangying@syau.edu.cn
Zheng Jinyu 2 3
Qi Hua 1 qihua10@syau.edu.cn
1 College of Agronomy, Shenyang Agricultural University , Shenyang , China
2 Institute of Agricultural Resources and Environments, Jilin Academy of Agricultural Sciences , Changchun , China
3 Key Laboratory of Crop Ecophysiology and Farming System in Northeast China, Ministry of Agriculture and Rural Affairs , Changchun , China
4 College of Agronomy and Biotechnology, China Agricultural University , Beijing , China
Adhikari Kabindra
Electronic publication date: 2022 May 13
Publication date: 2022
Volume: 10
Electronic Location ID: e13462
Received 2021 Feb 9; Accepted 2022 Apr 28
Copyright: © 2022 Sui et al.
Copyright year: 2022
Copyright holder: Sui et al.
License: This is an open access article distributed under the terms of the Creative Commons Attribution License, which permits unrestricted use, distribution, reproduction and adaptation in any medium and for any purpose provided that it is properly attributed. For attribution, the original author(s), title, publication source (PeerJ) and either DOI or URL of the article must be cited.
License URL: https://creativecommons.org/licenses/by/4.0/

Keywords: Tillage practices, Straw incorporation, Soil enzyme activities, Microbial communities, N fertilization

Funding: National Natural Science Foundation of China 32071976, 31901471 National Key Research and Development Program of China 2016YFD0300103 China Postdoctoral Science Foundation 2019M661130 Science and Technology Program in Liaoning Province of China 2019JH2/10200004 This work was supported by the National Natural Science Foundation of China (32071976, 31901471), the National Key Research and Development Program of China (2016YFD0300103), the China Postdoctoral Science Foundation (2019M661130), and the Science and Technology Program in Liaoning Province of China (2019JH2/10200004). The funders had no role in study design, data collection and analysis, decision to publish, or preparation of the manuscript.

==============================
Soil enzymes and microorganisms are both important to maintaining good soil quality and are also sensitive to changes in agricultural management. The individual effects of tillage, straw incorporation and nitrogen (N) fertilization on soil enzymes and microflora have been widely acknowledged, but their interactive effect remains largely unknown. In a 5–year in–situ field study, effects of rotary (RTS) and plow tillage (PTS) practices with straw incorporation combined with three N fertilization levels (0 kg N ha–1, CK; 187 kg N ha–1, MN; 337 kg N ha–1, HN) on soil enzyme activities and microbial communities were assessed. Our results showed that the activities of β–glucosidase (βG), N–acetylglucosaminidase (NAG) and acid phosphatase (APH) were improved in RTS+MN. The bacterial and fungal abundances in RTS+MN and RTS+HN were 1.27–27.51 times higher than those in other treatment groups. However, the bacterial and fungal alpha diversities were enhanced in PTS+MN and PTS+CK compared with other treatments, respectively. Proteobacteria and Basidiomycota were the predominant phylum for the respective bacterial and fungal communities. Moreover, significant interactive effects were found in the fungal community composition, but only minor impacts were observed on the bacterial community composition. Soil water content and penetration resistance contributed more to the soil enzyme activity and microbial community than other soil properties investigated, whereas there was a significant positive correlation between βG and APH activities and microbial abundance. These findings can provide new insights into tillage with straw incorporation and N fertilization on maize cultivation in northeast China.

Introduction

Soil microorganisms are important participants in assessing agricultural soil quality and ecosystem function through their integral and unique roles in mediating the decomposition of soil organic matter and the cycling of nutrients (Essel et al., 2019; Wang et al., 2020a). Soil enzymes catalyze the chemical reactions during the degradation processes of microorganisms leading to the subsequent release of nutrients into the soil (Li et al., 2019; Zhao et al., 2016a). Therefore, evaluating the relationships among the soil environment factors, enzyme activities, and microbial community is a critical step in developing sustainable agriculture.

Returning crop straw to fields is a sustainable approach that provides nutrients for crop growth and a carbon (C) source for soil microbe reproduction in agricultural systems (Li et al., 2019). This method has been proven to increase microbial biomass and enzyme activity compared with removing the crop straw (Staley, 1999; Sarker et al., 2019). Furthermore, the location of the straw incorporated into the soil is determined through tillage practices, which changes the physicochemical properties in soil and affects the soil residue degradation processes (Yang et al., 2016; Sarker et al., 2019). For example, rotary tillage employed for straw incorporation would further break the residues and mix them more thoroughly with soil, giving the straw easy contact with the surface soil and allowing it to quickly decompose (Helgason et al., 2014). Plow tillage for straw incorporation breaks the plow pan layers and buries the straw 25.0–30.0 cm deep (Mu et al., 2016). Deep plowing enhances precipitation interception and soil ventilation conditions, which is beneficial for microbial growth in deeper soils as well as increased enzyme activity to enhance the cycling of soil nutrients (Schneider et al., 2017; Essel et al., 2019). When straw is incorporated into the soil, bacteria decompose the most and aerobic microorganisms become predominant (Nicolardot et al., 2007; Xia et al., 2020). Existing studies on soil microbial diversity and community composition affected by conventional rotary and plow tillage practices do not evaluate the impact of straw incorporation in northeast China.

The application of nitrogen (N) fertilizer to agroecosystems usually increases soil organic matter, available nutrients, and thus improves crop yields (Zhao et al., 2016b). However, the excessive use of mineral N fertilizer can decrease the soil cation exchange capacity and pH (Russell, Laird & Mallarino, 2006; Cai et al., 2015), which can cause a variety of environmental problems such as groundwater contamination by nitrate leaching (De Paz & Ramos, 2004) and serious greenhouse gas emissions (Ma et al., 2010; Huang et al., 2017). The responses of soil microbial diversity and community structure to different durations and amounts of N application have been reported across a series of agroecosystems (Zeng et al., 2016; Yu et al., 2019). For example, N addition strongly reduces total microbial biomass and soil microbial activity (Ramirez, Craine & Fierer, 2012; Yu et al., 2019), and the relative abundance of copiotrophic microbial groups are also enhanced by N addition, whereas the amount of oligotrophic microbial groups decreases with N fertilization (Ramirez, Craine & Fierer, 2012; Li et al., 2019). Due to a higher C/N ratio of crop straw, such as maize straw, relative to the soil microbial biomass, N availability affects the microbial decomposition of crop straw (Zang, Wang & Kuzyakov, 2016; Li et al., 2017a). The effect of soil mineral N levels on crop straw and soil organic matter (SOM) decomposition varies depending on the levels of N applied and on the microbial community structure of the soil (Treseder, 2008; Kuzyakov & Xu, 2013). However, the interactive effects of tillage with straw incorporation and N fertilizer on soil microbial diversity and community composition are still not fully understood, so we sought to bridge this knowledge gap.

Northeast China is one of the major areas of agricultural production, with the region’s total maize yield accounting for approximately 30% of the nation’s total maize production (Liu et al., 2012). Maize straw production in this region totaled 72.3 million tons in 2014 accounting for 36.3% of the national yield (Li et al., 2017b). Promoting the incorporation of straw back into the field as fertilizer could decrease the use of chemical fertilizer and the burden of air pollution in this region (Yin et al., 2018). In this study, we established a field experiment consisting of rotary tillage and plow tillage for straw incorporation with three levels of N fertilization. The objectives were: (a) to evaluate the effect of tillage with straw incorporation and N levels on soil environment factors and enzyme activities; (b) to compare the responses of microbial abundance, diversity and community composition to the combined effects of tillage practices with straw incorporation and different N application levels; and (c) to explore the relationships between soil microbial communities, enzyme activities and environmental factors in northeast China for spring maize production.

Materials and Methods

Site description

This in–situ field experiment was done in 2015 in Tieling City (42°49′N, 124°16′E), Liaoning Province, China. Maize is the dominant crop in this area and is harvested once per year. The climate in this area is subtropical arid with a mean annual temperature of 20.9 °C and average precipitation of 543.0 mm during the spring maize growth period (from the beginning of May to the end of September). Before the field experiment started in April 2015, the basic soil chemical properties of the top 0–20 cm soil layer were: 15.7 g kg–1 of soil organic matter (SOM), 1.2 g kg–1 of total nitrogen (TN), 25.7 mg kg–1 of available phosphorus (AP), 109.3 mg kg–1 of available potassium (AK) and pH of 5.64.

Experimental design

This field experiment was performed using a split–plot design with three replicates. Rotary (RTS) and plow (PTS) tillage with straw incorporation were applied to the main plots. The three N fertilization levels were: 0 kg N ha–1 (control, CK), 187 kg N ha–1 (medium N application, MN) and 337 kg N ha–1 (high N application, HN) and were applied to the subplots. We designed five different nitrogen fertilizer gradients including 0, 112, 187, 262 and 337 kg N ha−1, it has been found in previous studies that maize grain yield would not increase significantly when N application more than 187 kg N ha−1 (Sui et al., 2020). Therefore, we chose 0, 187 and 337 kg N ha−1 in this research, represently. The main plot size was 28.8 m × 10 m and each subplot was 9.6 m × 10 m with the field plots arranged as shown in Fig. 1.

Figure 1 A schematic diagram of the experiment design and field arrangement.

After the maize harvest every year, maize straw was chopped into segments 5–10 cm in length. For RTS treatment, a rotary tiller mixed the soil twice to incorporate maize straw into the 0–15 cm soil layer. For PTS treatment, the soil was inverted with a plow tiller to bury the maize straw beneath the 30 cm soil layer. Basal fertilizer with 90 kg P2O5 ha–1 (Superphosphate) and 90 kg K2O ha–1 (potassium chloride) was used when the maize was planted. The N fertilizer (urea) was applied as split doses: 1/3 of the total N was basal fertilizer and the remainder was jointing fertilizer. Spring maize (Zea mays L, Zhengdan 958) was manually sown in early May and harvested at the end of September. Maize was planted at a density of 67,500 ha–1 with 60 cm interspaces.

Soil sampling and analysis of soil physicochemical properties

Soil samples were collected from the upper 30 cm of the soil at the maize jointing stage (June 2019). Five soil cores were randomly taken from three sampling sites in each plot and mixed to obtain three representative samples for each treatment. A portion of the soil sample was air–dried for soil property analysis. Subsamples were passed through a 2 mm sieve and stored at 4 °C and –80 °C for enzyme activity and molecular analyses, respectively.

Soil bulk density (BD) was measured using the the core ring method (Blake & Hartage, 1986). Soil water content (SWC) was determined by oven drying to a constant mass at 105 °C. Soil pH was assessed with a PHSJ–3F digital pH meter. The penetration resistance (PR) was measured using the SC900 (Field Scout, Portland, OR, USA). SOC and TN were determined using an elemental analyzer (EA 3000, Turin, Italy). Total phosphorus (TP) and total potassium (TK) were assessed by digestion and then spectrophotometer detection and flame photometer detection (Olsen, Sommers & Page, 1982). Nitrate (NO3––N) and ammonium (NH4+–N) were extracted with 2 M KCl for 1 h and determined colorimetrically using a Smart Chem 200 auto discrete analyzer according to the method described by Joseph & Henry (2008). AP and AK were determined using the sodium bicarbonate Olsen method (Nafiu, 2006) and the flame photometric method (Motsara & Roy, 2008), respectively.

Soil enzyme activities

Soil β–glucosidase (βG), N–acetylglucosaminidase (NAG), leucine aminopeptidase (LAP) and acid phosphatase (APH) activities were measured based on the method used by Saiya-Cork, Sinsabaugh & Zak (2002). We conducted assays using 96–well black microplates with eight replicate wells per sample, and the analysis also involved a blank, a negative control, and a quench standard. The microplates were incubated at 20 °C for 4 h in the dark. Fluorescence values were read by the Synergy H4 Hybrid Reader (SynergyH4 BioTek, Winooski, VT, USA) using a microplate fluorometer with 365 nm excitation and 450 nm emission filters. After accounting for negative controls and quenching, activities were expressed in units of nmol g–1 h–1.

DNA extraction, PCR amplification, and sequencing

The soil microbial DNA of each sample was extracted from 0.5 g frozen soil using a PowerSoil DNA Isolation Kit (MOBIO Laboratories, Carlsbad, CA, USA) according to the manufacturer’s instructions. The extracted DNA was checked on 1.2% agarose gel and the DNA quality and concentration were evaluated using the NanoDrop 2000 Spectrophotometer (Thermo Scientific, Wilmington, DE, USA).

The primer pair 338F (5′–ACTCCTACGGGAGGCAGCA–3′) and 806R (5′–GGA CTACHVGGGTWTCTAAT–3′) targeted the V3–V4 region of the bacterial 16S rRNA gene (Wang et al., 2015). The ITS5F (5′–GGAAGTAAAAGTCGTA ACAAGG–3′) and ITS2R primers (5′–GCTGCGTTCTTCATCGATGC–3′) (Duan et al., 2019) were used to amplify the fungal internal transcribed spacer (ITS1) region. The PCR components contained 5 μl reaction buffer (5×), 5 μl GC buffer (5×), 2 μl of 2.5 mmol L−1 dNTP, 1 μl of 10 μmol L−1 forward primer, 1 μl of 10 μmol L−1 reverse primer, 1 μl of DNA template, 9.75 μl of ddH2O and 0.25 μl of Q5 DNA polymerase (New England Biolabs, Beijing, China). The PCR process consisted of a 5 min initial denaturation step at 98 °C, followed by 25 or 28 cycles of 30 s at 98 °C, 30 s of annealing at 52 °C, and extension at 72 °C for 1 min, with a final extension at 72 °C for 5 min for bacteria and fungi, respectively. The PCR products were checked on 2% agarose gel and further purified using the TruSeq Nano DNA LT Library Prep Kit (Illumina, CA, USA). Subsequently, the purified amplicons were sequenced (468 bp for bacteria and 280 bp for fungi) according to an equimolar and paired–end method with the Illumina MiSeq platform (Majorbio Bioinformatics Technology, Shanghai, China). The PCR products of the 16S rRNA and ITS1 genes were used as a template for establishing the sequencing library using the Illumina Miseq Platform.

The raw sequence data was processed using the QIIME2 bioinformatics software. The very first step applied the DADA2 pipeline for de-noising and de-replicating of the paired-end sequences including chimera removal at the end (Callahan et al., 2016). The gene sequencing of 16S rRNA resulted in a total of 1,290,394 sequence reads. After filtering, 1,234,878 sequence reads remained, and 1,064,843 sequence reads remained after the quality control step. The sequence reads from the gene sequencing of ITS1 totaled 1,016,644, with 969,667 and 950,213 sequence reads remaining after filtering and quality control, respectively. Operational Taxonomic Units (OTUs) were clustered at a 97% sequence identity cut–off using the QIIME software (Version 1.17). The effective sequences obtained from pyrosequencing were compared with Greengenes 16S rRNA and UNITE ITS gene database using NCBI’s BLASTN tool, and the species distribution diagram was employed. The taxonomic classification of OTUs was based on the Ribosomal Database Project classifier according to a 70% confidence threshold. The bacterial and fungal alpha–diversity, including the richness estimator of the Chao1 index and the diversity indices of Shannon index, were also analyzed.

Quantitative PCR (qPCR)

Real–time PCR amplification of the bacterial 16S rRNA and fungal ITS1 genes were performed on the CFX96 (Bio–Rad, Hercules, CA, USA) using the AceQ qPCR SYBR Green Master Mix (Jizhenbio, Shanghai, China). The reaction mixtures contained 7.5 μl SYBR Green Mix (2×) for each primer and 1 μl of DNA extract as the template. The amplification profile was 95 °C (5 min), followed by 40 cycles of 95 °C (10 s), 55 °C (15 s), 75 °C (30 s), and 1 cycle of 95 °C (15 s), 60 °C (1 min), 95 °C (45 s) for bacteria and fungi. Cycle threshold (Ct) values were determined with the Bio–Rad CFX Manager software (Bio–Rad, Hercules, CA, USA). Standard curves were generated by serially diluting plasmids, as described by Wang et al. (2020b). At the end of each PCR run, a melting curve analysis was performed to evaluate the amplification specificity (Yu et al., 2019). The amplification efficiency was 91.8% and 105.2% and R2 value was 0.991, 0.994 for the bacterial and fungal communities, respectively.

Statistical analysis

The differences between the effects of tillage with straw incorporation or N levels and their interaction on soil properties, enzyme activities, alpha diversity indices and abundance of bacterial and fungal were tested using two factor analysis of variance (ANOVA) and Duncan’s multiple range Test with SPSS 23.0 (SPSS Inc. Chicago, IL, USA). A Spearman’s correlation analysis was performed to reveal the relationships between soil properties, enzyme activities, gene abundance and the bacterial and fungal alpha diversity indices. The clustering of different samples and the microbial community structure was illustrated by nonmetric multidimensional scaling (NMDS) as microbial beta diversity. Correlations among the soil properties and microbial compositions were determined using a redundancy analysis (RDA) in CANOCO 4.5. The figures were drawn in Origin 9.0. A p value < 0.05 was considered statistically significant.

Results

Variations in the physicochemical properties of the soil and the enzyme activities after tillage with straw incorporation and N fertilization

Tillage with straw incorporation (Ts), N fertilization (N) and their interaction (Ts × N) greatly affected soil properties and enzyme activities (Table 1). The addition of N decreased soil pH in both RTS and PTS treatments. The SOC, TN, NO3––N and AP contents under RTS treatment increased with N addition, and NH4+–N, AK and TP contents under PTS treatment were also greater with increasing N levels. However, SOC and TN contents decreased in PTS treatment (Table 2 and Fig. 2). Averaged over all N levels, SWC, BD, PR, SOC, TN, AP and AK contents of RTS treatment were higher than in the PTS treatment, but pH, TP, TK, NO3––N and NH4+–N contents under RTS were lower compared to PTS (Table 2 and Fig. 2; p < 0.05).

Table 1 The ANOVA F-values of the effects of tillage with straw incorporation and N fertilization on soil physicochemical properties, enzyme activities, and bacterial and fungal gene abundance and diversity.

Varieties	Properties	Tillage with straw incorporation (Ts)	N levels (N)	Ts × N	
Soil physicochemical properties	pH	573.29***	649.93***	6.50*	
Soil water content (SWC)	46.92***	13.74**	4.38*	
Soil bulk density (BD)	149.76***	12.83**	3.58ns	
Penetration resistance (PR)	103.65***	11.02**	1.46ns	
Soil organic carbon (SOC)	42.62***	7.01*	24.21***	
Total nitrogen (TN)	45.43***	1.74ns	18.43***	
Total phosphorus (TP)	31.15***	38.95***	36.18***	
Total potassium (TK)	80.41***	127.69***	101.98***	
Ammonium (NH4+-N)	19.42**	6.79*	11.04**	
Nitrate (NO3--N)	166.59***	119.36***	64.66***	
Available phosphorus (AP)	86.01***	133.95***	10.36**	
Available potassium (AK)	900.00***	694.78***	190.33***	
Enzyme actives	β-glucosidase (βG)	196.52***	142.51***	71.46***	
N-acetylglucosaminidase (NAG)	51.88***	57.05***	21.79***	
Leucine aminopeptidase (LAP)	1.48ns	6.16*	1.01ns	
Acid phosphatase (APH)	75.41***	59.97***	17.81***	
Bacterial	Bacterial gene copy numbers (BGCN)	249.42***	1.64ns	25.64***	
Bacterial Chao 1 index (BCI)	13.85**	0.30ns	2.65ns	
Bacterial Shannon index (BSI)	17.05**	1.00ns	2.05ns	
Fungal	Fungal gene copy numbers (FGCN)	233.06***	51.68***	44.49***	
Fungal Chao 1 index (FCI)	4.92*	10.85**	4.47*	
Fungal Shannon index (FSI)	33.33***	141.49***	37.81***	
Note:

Ns, no significant difference; One (*), two (**) and three (***) asterisks indicate a significant difference among treatments at p < 0.05, p < 0.01 and p < 0.001, respectively.

Table 2 Soil properties under tillage with straw incorporation practices and N fertilization.

Treatment	pH	SWC (%)	BD (g cm−3)	PR (Mpa)	
RTS	CK	5.67 ± 0.01b	18.09 ± 0.07bc	1.39 ± 0.04b	397.64 ± 30.12b	
MN	5.31 ± 0.02e	18.93 ± 0.25b	1.44 ± 0.04a	527.74 ± 29.88a	
HN	5.13 ± 0.02f	20.41 ± 1.14a	1.41 ± 0.03ab	522.23 ± 58.95a	
PTS	CK	5.88 ± 0.03a	16.87 ± 0.14d	1.30 ± 0.05d	222.46 ± 46.29d	
MN	5.44 ± 0.02d	17.82 ± 0.53cd	1.35 ± 0.04c	268.28 ± 24.96cd	
HN	5.60 ± 0.04c	17.68 ± 0.09cd	1.28 ± 0.06d	324.54 ± 65.42bc	
Note:

Rotary tillage with straw incorporation (RTS), Plow tillage with straw incorporation (PTS), 0 (CK), 187 (MN) and 337 (HN) kg N ha−1 applied. Soil water content (SWC), soil bulk density (BD), penetration resistance (PR). The values are mean ± standard deviation (n = 3). Different letters indicate comparisons with significant difference (p < 0.05) between treatments.

Figure 2 Changes in total soil nutrients (A) and available nutrients (B) after tillage with straw incorporation practices and N fertilization levels.

Different letters indicate significant differences (p < 0.05) between treatments. The values are mean ± standard deviation (n = 3). Soil organic carbon (SOC), total nitrogen (TN), total phosphorus (TP), total potassium (TK), nitrate (NO3−-N), ammonium (NH4+-N), available phosphorus (AP), and available potassium (AK). Rotary tillage with straw incorporation (RTS), Plow tillage with straw incorporation (PTS), 0 (CK), 187 (MN) and 337 (HN) kg N ha−1 applied.

As for soil enzyme activity, the activities of βG, NAG and APH were significantly enhanced by N addition in RTS treatments (Fig. 3). However, compared with CK, MN increased the activity of βG, NAG and APH, while HN decreased the activity of βG and NAG in PTS treatments. Moreover, RTS treatments led to significant increases in βG, NAG and APH activity of 33.3%, 37.1% and 23.3%, respectively, but no remarkable variation was observed for LAP compared with PTS treatments (p < 0.05). Furthermore, the activities of βG, NAG, LAP and APH were positively related to soil AP and AK contents, and negatively related to pH (Table 3). Meanwhile, the activity of βG, NAG and APH were also positively associated with SWC, BD, PR, SOC and TN (p < 0.05).

Figure 3 Changes in soil enzyme activities after tillage with straw incorporation and N fertilization.

Different letters indicate significant differences (p < 0.05) between treatments. The values are mean ± standard deviation (n = 3). β-glucosidase (βG), N-acetylglucosaminidase (NAG), leucine aminopeptidase (LAP), acid phosphatase (APH). Rotary tillage with straw incorporation (RTS), Plow tillage with straw incorporation (PTS), 0 (CK), 187 (MN) and 337 (HN) kg N ha−1 applied.

Table 3 Pearson’s correlation coefficients between enzyme activities and soil properties.

Soil properties	βG	NAG	LAP	APH	
pH	–0.818**	–0.731**	–0.531*	–0.875**	
SWC	0.616**	0.494*	0.242	0.607**	
BD	0.803**	0.794**	0.349	0.783**	
PR	0.614**	0.491*	0.228	0.657**	
SOC	0.770**	0.805**	–0.004	0.723**	
TN	0.773**	0.749**	–0.062	0.741**	
TP	0.172	0.153	0.323	0.125	
TK	0.171	0.231	0.438	0.114	
NH4+-N	–0.293	–0.226	0.426	–0.199	
NO3--N	0.135	0.220	0.306	0.078	
AP	0.900**	0.854**	0.599**	0.894**	
AK	0.609**	0.563*	0.698**	0.685**	
Note:

Soil water content (SWC), soil bulk density (BD), penetration resistance (PR), soil organic carbon (SOC), total nitrogen (TN), total phosphorus (TP), total potassium (TK), ammonium (NH4+–N), nitrate (NO3––N), available phosphorus (AP), available potassium (AK), β–glucosidase (βG), N–acetylglucosaminidase (NAG), leucine aminopeptidase (LAP), acid phosphatase (APH). One (*) and two (**) asterisks indicate significant difference among treatments at p < 0.05 and p < 0.01, respectively.

Effect of tillage with straw incorporation and N fertilization on microbial abundance

Ts and its interaction effect with N had significant effects on the bacterial and fungal gene abundance of the soil (Table 1). N addition significantly increased bacterial (Fig. 4A) and fungal (Fig. 4D) gene copy numbers in RTS treatments, however, for PTS treatments, decreased trends were observed with N addition (Fig. 4A). Moreover, RTS treatments increased bacterial (Fig. 4A) and fungal (Fig. 4D) gene copy numbers compared with PTS treatments. The highest and lowest bacterial gene copy numbers (ranging from 3.0 × 105 to 2.5 × 106 copies g−1 soil) were detected in RTS+MN treatment and PTS+MN treatment, respectively. The highest fungal gene copy number (1.5 × 106 copies per g−1 of soil) was in RTS+HN treatment and the lowest (4.1 × 104 copies per g−1 of soil) was in PTS+MN treatment.

Figure 4 Changes in soil microbial (bacterial and fungal) abundance (gene copy numbers) and alpha diversity (Chao 1 and Shannon index) after tillage with straw incorporation and N fertilization.

Different letters indicate significant differences (p < 0.05) between treatments. The values are mean ± standard deviation (n = 3). Rotary tillage with straw incorporation (RTS), plow tillage with straw incorporation (PTS), 0 (CK), 187 (MN) and 337 (HN) kg N ha−1 applied.

The Pearson’s correlation coefficients between soil properties and enzyme activities, as well as microbial abundance, showed that the bacterial gene copy number was significantly related to soil pH, SWC, BD, PR, SOC, TN, TP, TK, NH4+–N, NO3––N, βG, NAG and APH (Fig. 5, p < 0.05). The fungal gene copy number was significantly related to soil pH, SWC, BD, PR, SOC, TN, NH4+–N, βG and APH (p < 0.05).

Figure 5 Pearson’s correlation coefficients between the soil properties, enzyme activities and microbial abundance and diversity.

One asterisk (*) and two asterisks (**) indicate the correlation is significant at the 0.05 and 0.01 level, respectively. Soil water content (SWC), soil bulk density (BD), penetration resistance (PR), soil organic carbon (SOC), total nitrogen (TN), total phosphorus (TP), total potassium (TK), nitrate (NO3−-N), ammonium (NH4+-N), available phosphorus (AP), available potassium (AK), β-glucosidase (βG), N-acetylglucosaminidase (NAG), leucine aminopeptidase (LAP), acid phosphatase (APH), bacterial gene copy numbers (BGCN), bacterial Chao 1 index (BCI), bacterial Shannon index (BSI), fungal gene copy numbers (FGCN), fungal Chao 1 index (FCI), fungal Shannon index (FSI).

Effect of tillage with straw incorporation and N fertilization on microbial diversity

Ts had a large influence on the soil bacterial and fungal alpha diversity, N and Ts × N greatly affected soil fungal alpha diversity (Table 1). Overall, significant effects on Chao 1 and the Shannon index were found between treatments from both bacteria and fungi (p < 0.05), but greater variations were obviously observed from the fungal diversity indices compared to the response of bacteria on treatments (Figs. 4B, 4C, 4E, 4F). Moreover, treatments of increasing N input markedly lessened the Shannon diversity index of fungi compared with CK treatments with either RTS or PTS (Fig. 4F). Similar trends were also seen in the Chao 1 index of fungi, but with slight variations among treatments.

The results of the Pearson’s correlation coefficient tests showed that the bacterial Chao 1 index was significantly related to SWC, PR, TP, TK, NH4+–N and NO3––N (Fig. 5, p < 0.05). Similarly, the bacterial Shannon index was significantly related to pH, SWC, BD, PR, SOC, TN, NH4+–N, AP, AK, βG, NAG and APH (p < 0.05). However, the fungal Chao 1 index was significantly related to SWC, PR and TN (p < 0.05). Similarly, the fungal Shannon index was significantly associated with pH, SWC, PR and TN (p < 0.05).

A nonmetric multidimensional scaling analysis (NMDS) reflecting microbial beta diversity indicated that there was no significant changes in the number of bacteria observed (Fig. 6). With respect to fungi, the diversities from RTS+MN, PTS+MN and PTS+HN treatments were nearly the same, whereas the RTS+CK, RTS+HN and PTS+CK treatments were greatly affected by Ts and N level treatments.

Figure 6 Nonmetric multidimensional scaling analysis (NMDS) of changes in soil bacterial and fungal beta diversity after tillage with straw incorporation and N fertilization levels.

Rotary tillage with straw incorporation (RTS), plow tillage with straw incorporation (PTS), 0 (CK), 187 (MN) and 337 (HN) kg N ha−1 applied.

Effect of tillage with straw incorporation and N fertilization on microbial composition

According to bacterial community composition, the dominant phyla (abundance >1%) across all treatments were Proteobacteria, Actinobacteria, Acidobacteria, unclassified_Bacteria, Bacteroidetes, Gemmatimonadetes, Chloroflexi and Patescibacteria, with contributions of 36.65%, 27.48%, 12.09%, 7.74%, 4.45%, 4.43%, 2.90% and 1.46%, respectively (Fig. 7A). Among them, a relatively higher abundance of Proteobacteria, Acidobacteria, Chloroflexi and Patescibacteria were observed under RTS treatments compared with PTS treatments. In contrast, lower relative abundance of Actinobacteria and unclassified_Bacteria were observed under RTS treatments relative to PTS treatments. Similarly, such trends of taxonomic composition were observed at the class and order level as well (Tables S1 and S3). Among all sequences, the dominant fungal phyla were Basidiomycota, Ascomycota, unclassified_Fungi and Mortierellomycota, with average contributions of 63.32%, 29.23%, 2.81% and 2.02%, respectively (Fig. 7B). Among them, a relatively greater abundance of Basidiomycota and smaller relative abundance of Ascomycota and Mortierellomycota were present under RTS treatments compared to PTS treatments. Additionally, increasing levels of N reduced the relative abundance of Ascomycota, unclassified_Fungi and Mortierellomycota, but improved the relative abundance of Basidiomycota, compared with CK treatments. Similar results between treatments were also observed at the class and order level of microbial composition (Tables S2 and S4).

Figure 7 Changes in soil microbial (bacterial and fungal) taxonomic composition at the phylum level after tillage with straw incorporation and N fertilization levels.

Rotary tillage with straw incorporation (RTS), Plow tillage with straw incorporation (PTS), 0 (CK), 187 (MN) and 337 (HN) kg N ha−1 applied. The groups accounting for 1% are shown, whereas those accounting for <1% are combined as Others.

The bacterial and fungal taxa that responded to soil property changes at the phylum taxonomic level are shown in Fig. 8. SWC, PR, BD, SOC, TN and AK were considerably related to changes in Proteobacteria, Actinobacteria, Acidobacteria, unclassified_Bacteri and Chloroflexi for bacterial phyla (Fig. 8A). Furthermore, soil properties such as the pH, SWC and PR were considerably related to changes in the relative abundances of the fungal phyla Basidiomycota, Ascomycota and Mortierellomycota (Fig. 8B).

Figure 8 Ordination plots of the results from the redundancy analysis (RDA) to identify the relationships between the microbial (bacterial and fungal) taxa (blue arrows) and the soil properties (red arrows) at the phylum level.

(A) The relationship between the soil bacterial taxa and the soil properties; (B) the relationship between the soil fungal taxa and the soil properties. Bacterial taxa: Proteobacteria (Prot), Actinobacteria (Acti), Acidobacteria (Acid), unclassified_Bacteria (uncl-B), Bacteroidetes (Bact), Gemmatimonadetes (Gemm), Chloroflexi (Chlo), Patescibacteria (Pate), Verrucomicrobia (Verr). Fungal taxa: Basidiomycota (Basi), Ascomycota (Asco), unclassified_Fungi (uncl-F), Mortierellomycota (Mort). Soil properties: pH, soil water content (SWC), soil bulk density (BD), penetration resistance (PR), soil organic carbon (SOC), total nitrogen (TN), total phosphorus (TP), total potassium (TK), nitrate (NO3−-N), ammonium (NH4+-N), available phosphorus (AP), available potassium (AK).

Discussion

Responses of soil properties and enzyme activities to tillage with straw incorporation and N fertilization

Sustainable crop productivity is defined by the changes in soil physicochemical parameters associated with soil quality (Karlen et al., 1997). Xue et al. (2015) reported that rotary tillage with straw incorporation significantly increased surface layer (0–30 cm depth) SOC and TN concentrations, which was also reflected in our study. AP and AK contents (Fig. 2) were significantly higher and corresponded with SOC and TN concentration trends compared with PTS treatments. These results indicated that the fields under RTS treatments were rotated to 15 cm depth, thereby uniformly distributing the straw in the 0–15 cm soil layer. Straw was mixed into the topsoil, which accelerated straw decomposition and SOC and TN immobilization (Dikgwatlhe et al., 2014; Helgason et al., 2014). Shallow tillage with straw improved soil microbial biomass and increased the proportion of microbial biomass C in total SOC (Balota et al., 2003; Heinze, Rauber & Joergensen, 2010). Furthermore, rotary tillage caused a plow pan layer and decreased SOM decomposition at 15–30 cm soil depth (Kabiri, Raiesi & Ghazavi, 2016). However, around the jointing stage of maize, the decomposition of straw and SOM released minerals, which could be acquired by microorganisms, then mineralized to NH4+ or NO3– and were likely utilized by crop roots (Kuzyakov & Xu, 2013). Thus, most of the N fertilization is allocated to microorganisms shortly after N application or released from decomposing straw (Kuzyakov & Xu, 2013; Li et al., 2017a). As a result, lower NH4+–N and NO3––N content and higher AP and AK content were found in RTS treatments. Conversely, straw was buried into the 25–30 cm soil layers in PTS treatments, and the favorable soil ventilation conditions and broken soil aggregates with acquired straw promoted SOM decomposition and CO2 emissions in the 0–20 cm soil layer (Dong et al., 2008). Thus, lower SOC and TN contents were found in PTS treatments. In addition, changes in SOC and TN under different tillage treatments were affected by N levels in this study. SOC and TN contents were enhanced with N addition in RTS treatments, whereas the opposite results were observed in PTS treatments. This might be because N addition improved soil labile SOC and TN pool in RTS and promoted native soil SOC and TN mineralization in PTS (Sommer et al., 2011; Pu et al., 2019). Furthermore, a higher available nutrient content was present with MN under both RTS and PTS treatments. These results reveal that MN application level could be considered as an optimal N application level and contribute to soil microbial reproduction and nutrient supply.

Soil enzymes played a vital role in the cycling of soil nutrients in agroecosystems, and soil enzyme activity could be used as an index of soil microbial activity (Zhao et al., 2016a). Previous studies reported that the effect of shallow tillage with straw increased soil enzyme activities in the 0–20 cm soil layer, including β–glucosidase, urease, phosphatase and catalase activities (López-Garrido et al., 2014; Kabiri, Raiesi & Ghazavi, 2016). Therefore, it is reasonable that significant improvements in soil βG, NAG and APH activities were also observed in RTS treatments (Fig. 3), suggesting that straw mixed with surface soil could enhance soil enzyme secretion, which is critical for straw decomposition and SOC sequestration. Meanwhile, higher soil βG, NAG, LAP and APH activities were shown at the MN application level. Therefore, the increase in soil available nutrient contents with MN treatment appeared to be a consequence of higher soil enzyme activity for nutrient cycling. Furthermore, our research displayed that the activities of four soil enzymes were significantly correlated with the physical and chemical properties of soil (Table 3). Higher soil enzyme activities corresponded to an increase in soil nutrient accumulation (SOC, TN, AP and AK), which was likely due to positive feedback. Straw and SOM contain sufficient substrates to stimulate the synthesis of these enzymes (Kabiri, Raiesi & Ghazavi, 2016). There was also a positive relationship between soil enzyme activities and physical properties (e.g., BD, PR and SWC). This suggests that tillage practices changed the location of straw incorporation into soil and affected soil structure, further regulating soil enzyme activities. In addition, there was a significant negative correlation between soil enzyme activities and soil pH, which was consistent with Dick, Rasmussen & Kerle (1988), who reported that long–term straw return practices and N fertilizer addition altered the soil pH, catalytic efficiency and soil enzyme activities.

Responses of microbial abundance and diversity to tillage with straw incorporation and N fertilization

Many believe that the incorporation of crop straw into soil might intensify SOM mineralization. This effect is named the ‘priming effect’ and has been clearly shown at the rhizosphere scale (Broadbent, 1947; Bingeman, Varner & Martin, 1953). In addition, linking experiments with modeling has revealed that, shortly after the input of substrates, the microbial turnover increases (termed the ‘apparent priming effect’), and only later does the turnover of SOM change a significant extent (the real priming effect) (Blagodatsky et al., 2010). Our results found that RTS treatments significantly increased the bacterial and fungal gene copy numbers compared with PTS treatments (Figs. 4A, 4D). Moreover, there were significant positive effects of SWC, BD, PR, SOC, TN, βG and APH on both bacterial and fungal gene levels (Fig. 5). These results indicated that higher SWC, BD and PR in RTS treatments were more incorporated into the soil and straw, which greatly stimulated the increase of microbial abundance, leading to soil microbial communities dominated by the fast–growing r–strategists (Chen et al., 2014). These microorganisms could produce extracellular enzymes (such as βG, which was responsible for degrading cellulose), facilitate the decomposition of crop straw, and increase the apparent priming effect and straw–derived CO2 emissions (Kabiri, Raiesi & Ghazavi, 2016; Chen et al., 2014). Conversely, PTS treatments buried the straw beneath the plow layer and the microbial communities in the topsoil hardly contacted the straw layers, leading to slow growing K–strategists becoming the dominant soil microbes (Chen et al., 2014). These microorganisms have advantages in utilizing recalcitrant organics and could elevate the real priming effect and SOM–derived CO2 emissions, thus decreasing SOC and TN content (Fontaine, Mariotti & Abbadie, 2003; Chen et al., 2014). Furthermore, there were significant negative effects of NH4+–N on bacterial and fungal gene abundance. These results demonstrated that RTS decreased NH4+–N due to microbial immobilization, and thus intensified the competition for NH4+–N between plants and microorganisms (Kuzyakov & Xu, 2013; Li et al., 2017a). However, N addition significantly impacted the growth of microbial communities under both Ts systems. Our results showed that higher bacterial abundance under RTS treatments was found after MN treatment, while fungal abundance significantly increased with N addition. These phenomena followed the ‘microbial stoichiometry’ theory (Hessen et al., 2004), which states that r–strategist microorganisms would markedly enhance with N application, and thus accelerate the degradation of the crop straw and improve SOC and TN content (Chen et al., 2014). On the other hand, in this study, the bacterial gene abundance in PTS treatments decreased with N addition, while fungal abundance was not significantly changed. These results are explained by the ‘microbial N mining’ theory, which demonstrated that K–strategist microorganisms multiply under low–N availability conditions (Moorhea & Sinsabaugh, 2006).

In this study, we observed that the Chao 1 and Shannon index of both bacteria and fungi were significantly influenced by the Ts factor (Table 1). On average, there was a relatively higher Chao 1 and Shannon index of both bacteria and fungi after PTS treatments (3,548.3, 350.3, 10.4 and 4.9) than RTS treatments (3,230.5, 317.3, 10.1 and 4.3; Fig. 4). In addition, there were significant negative effects of SWC and PR on bacterial and fungal alpha diversities (Fig. 5), indicating that PTS treatments might break the plow pan layers and improve soil ventilation conditions, thus decreasing SWC and PR and increasing microbial diversities (Schneider et al., 2017; Essel et al., 2019). Furthermore, there was a negative correlation between bacterial and fungal diversities and soil pH observed in our study, which was similar to observations from a previous investigation (Zhou et al., 2016). However, N addition has different effects on the alpha diversities of bacteria and fungi. Bacterial alpha diversities were slightly changed with N fertilization in this study (Fig. 4), while all soil samples with N addition had lower fungal alpha diversities and showed a trend towards a negative response to TN (Fig. 5), which was consistent with the result that N fertilizer application reduced the diversity of fungi (Zhou et al., 2016). They also suspected that the decrease in fungal diversities coincided with the inordinately high concentrations of TN in ultra–rich soils. Moreover, soil microbial beta diversity could represent the overall response of microbial community structure to treatments in this study. Different treatments in this study changed different characteristics of bacteria and fungi (Fig. 6); the fungi community compositions varied more than bacteria in response to the combined treatment effects.

Responses of microbial compositions to tillage with straw incorporation and N fertilization

Bacteria and fungi represent overall soil biodiversity and dominate the essential soil processes, so the soil nutrients content and structural traits of soil can be affected by changing the bacterial and fungal community composition (Bastian et al., 2009). Due to relatively high soil humidity and compact soil structure in RTS treatments, the amount of readily available straw C and nutrients are abundant for soil microbial growth, thus stimulating the growth of both the bacterial and fungal groups, including Proteobacteria, Acidobacteria and Basidiomycota at the phylum level (Fig. 7), Alphaproteobacteria, Actinobacteria, Acidobacteriia and Tremellomycetes at the class level (Tables S1, S2), and Unclassified–alphaproteobacteria, Acidobacteriales and Cystofilobasidiales at the order level (Tables S3, S4). Alphaproteobacteria is part of the copiotrophic bacterial group (r–strategists), which are adapted to break down the straw into simpler compounds in the early stages of decomposition (Li et al., 2019). It has been proposed that Acidobacteria, Basidiomycota and their respective classes (Acidobacteriia and Tremellomycetes) and orders (Acidobacteriales and Cystofilobasidiales) are oligotrophic microorganisms (K–strategists), involved in the degradation of complex substrates (cellulose, hemicelluloses and lignin; Bastian et al., 2009). It is also generally accepted that Actinobacteria contain both r–strategists and K–strategists (Morrissey et al., 2016). These microbial taxonomic groups colonize straw and then mineralize straw–derived C releasing CO2. In contrast, with the heavily ventilated soil conditions after PTS treatments, there are higher taxonomic divisions in Actinobacteria and Ascomycota at the phylum level (Fig. 7), Sordariomycetes at the class level (Table S2), and Betaproteobacteriales and Sordariales at the order level (Tables S3, S4). It is commonly believed that Ascomycota and its respective order, Sordariomycetes, and class, Sordariales, are r–strategists, which might affect the decomposition of soil organic matter (Xiong et al., 2014). Similarly, Actinobacteria and Betaproteobacteria have been described as r–strategists (Morrissey et al., 2016) or K–strategists (Ishii et al., 2011), but previous research observed an enrichment of such bacteria at the decomposition of complex and recalcitrant substrates (Li et al., 2019). In general, these microbial communities facilitated the mobilization of SOM, thus decreasing the SOC and TN content in this study. These results warrant further study for more insights on the benefits of returning crop straw to the field.

Fungal communities were markedly affected by N addition relative to bacterial communities (Fig. 7). Basidiomycota and its corresponding classes and orders were improved with N addition (Tables S2, S4), while the opposite results were observed in Ascomycata and its corresponding classes and orders. During straw decomposition, Basidiomycota played a role later in the degradation process with their ability to degrade recalcitrant organic matter. Ascomycata might play a crucial part in the initial steps of straw decomposition (Bastian et al., 2009). These results indicated that the combination of tillage with straw and N fertilizer significantly increased K–strategist fungal groups, which was beneficial for the degradation process of maize straw (Li et al., 2019).

Conclusions

This study has shown the different impacts of tillage with straw incorporation and N levels on soil enzyme activities and microbial communities from spring maize in northeast China. Our results demonstrate that rotary tillage with straw incorporation significantly increased soil enzyme activities and microbial (bacteria and fungi) abundance and decreased alpha diversity compared to plow tillage with straw incorporation. Increasing N input was not beneficial for fungal diversity, but a medium N application amount could improve soil enzyme activities. Moreover, the fungal community composition varied significantly relative to the community composition of bacteria in response to the combined effects of tillage with straw incorporation and applied N levels. Additionally, soil water content and penetration resistance played important roles in driving the soil enzyme activity, microbial abundance and community composition. Overall, these results suggest that rotary tillage with straw incorporation and medium N fertilizer application were helpful for the short–term improvement of soil properties and soil microbes in northeast China.

Supplemental Information

Supplemental Information 1 Changes in soil bacterial taxonomic composition relative abundance (%) at the class level under different tillage with straw incorporation and N levels.

Rotary tillage with straw incorporation (RTS), Plow tillage with straw incorporation (PTS), 0 (CK), 187 (MN) and 337 (HN) kg N ha–1 applied. The values are mean ± standard deviation (n = 3). The groups accounting for 1% are shown, whereas those accounting for <1% are combined (Others).

Click here for additional data file.

Supplemental Information 2 Changes in soil fungal taxonomic composition relative abundance (%) at the class level under different tillage with straw incorporation and N levels.

Rotary tillage with straw incorporation (RTS), Plow tillage with straw incorporation (PTS), 0 (CK), 187 (MN) and 337 (HN) kg N ha–1 applied. The values are mean ± standard deviation (n = 3). The groups accounting for 1% are shown, whereas those accounting for <1% are combined (Others).

Click here for additional data file.

Supplemental Information 3 Changes in soil bacterial taxonomic composition relative abundance (%) at the order level under different tillage with straw incorporation and N levels.

Rotary tillage with straw incorporation (RTS), Plow tillage with straw incorporation (PTS), 0 (CK), 187 (MN) and 337 (HN) kg N ha–1 applied. The values are mean ± standard deviation (n = 3). The groups accounting for 1% are shown, whereas those accounting for <1% are combined (Others).

Click here for additional data file.

Supplemental Information 4 Changes in soil fungal taxonomic composition relative abundance (%) at the order level under different tillage with straw incorporation and N levels.

Rotary tillage with straw incorporation (RTS), Plow tillage with straw incorporation (PTS), 0 (CK), 187 (MN) and 337 (HN) kg N ha–1 applied. The values are mean ± standard deviation (n = 3). The groups accounting for 1% are shown, whereas those accounting for <1% are combined (Others).

Click here for additional data file.

Additional Information and Declarations

Competing Interests

Author Contributions

Data Availability

The authors declare that they have no competing interests.

Pengxiang Sui conceived and designed the experiments, performed the experiments, analyzed the data, prepared figures and/or tables, authored or reviewed drafts of the paper, and approved the final draft.

Ping Tian performed the experiments, analyzed the data, prepared figures and/or tables, authored or reviewed drafts of the paper, and approved the final draft.

Zhengyu Wang performed the experiments, analyzed the data, prepared figures and/or tables, authored or reviewed drafts of the paper, and approved the final draft.

Hongli Lian performed the experiments, analyzed the data, prepared figures and/or tables, authored or reviewed drafts of the paper, and approved the final draft.

Yadong Yang analyzed the data, prepared figures and/or tables, authored or reviewed drafts of the paper, and approved the final draft.

Ziqi Ma performed the experiments, analyzed the data, prepared figures and/or tables, authored or reviewed drafts of the paper, and approved the final draft.

Ying Jiang conceived and designed the experiments, performed the experiments, analyzed the data, prepared figures and/or tables, authored or reviewed drafts of the paper, and approved the final draft.

Jinyu Zheng performed the experiments, analyzed the data, prepared figures and/or tables, authored or reviewed drafts of the paper, and approved the final draft.

Hua Qi conceived and designed the experiments, performed the experiments, analyzed the data, prepared figures and/or tables, authored or reviewed drafts of the paper, and approved the final draft.

The following information was supplied regarding data availability:

The data is available at NCBI SRA: PRJNA691700.

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
