# Peer review of "Soil properties and microbial communities of spring maize filed in response to tillage with straw incorporation and nitrogen fertilization in northeast China"

_PeerJ, doi:10.7717/peerj.13462_

## Round 0.1 · original submission · Major Revisions

The manuscript has been critically reviewed and based on the reports, a major revision is recommended. I also suggest considering a professional English edit.

Reviewer 1 ·

Basic reporting

The English language should be improved in all text.

Experimental design

The experiment design is not clearly, in my mind it is a two facters split–plot
design, and straw is not a factor. However, in introduction part, I feel clearly it is a three factors experiment. The experiment design should be clearly.

Validity of the findings

This study evaluates the effect of tillage with straw incorporation and N levels on soil environment factors, enzyme activities, microbial abundance, diversity and community composition and explore the relationships between soil microbial communities, enzyme activities and environment factors. It has some important data should be published after major revising.

Additional comments

Line 6:“researching the relationships....”is Chinese English, the English language should be better
Line 24-25: in fact, the soil microbial diversity and community composition has been evaluated such as Lu et al. 2018.
Line 64-66: The English language should be improved. And I think “study” is not good, experiment should be better.
Line 249-252: “However, at jointing stage of maize, decomposition of straw and SOM released minerals, such as amino acids and amino sugars, which could be acquired by microorganisms, then mineralized to NH4+ or NO3– , utilized by crop roots” is this true?
The discussion part should be improved.
You have many treatment, while you used “between” in all table title.

·

Basic reporting

This study reported the effects of tillage practices (rotary tillage and plow tillage) with straw incorporation with three N fertilization levels on soil microbial communities and enzyme activities. The language is of sufficient quality. However, the data analysis for the 16S rRNA sequencing result is not accurate and needed to be improved. For these details, please refer to the general comments.

Experimental design

The experimental design is clear and reasonable. However, there is no control group included. The best way is to add a control group without any treatments or without the application of straw. In that case, the data should be sufficient to reflect the effect of tillage practices and the straw incorporation.

Validity of the findings

The study is really helpful and supportive for the agriculture application and practices.

Additional comments

Line 7. The whole microbial community has a stronger relationship with these different factors, rather than only microbial diversity.

Line 9. A practical measure? Or approach?

Line 67, I suggested that you can draw a figure to show the plots design, which can indicate where is the main plots and where is the subplots, as well as which treatment on the different plots.

Line 124. It is accurate number for the amplicon length of bacteria based on 338F and 806R? Please correct it.

Lin196-197. You mentioned that PTS improved Chao1 and Shannon index in fungi, however, the figure 3E didn’t indicate the same result. Chao1 index also showed that there is no significant difference between RTS with CK and MN, and PTS. The description of fungi is not correct.

Line 198-199. There is no difference among PTS+CK, PTS+MN, PTS+HN, RTS+CK and RTS+MN in Fungal Chao 1 analysis. It is not correct to describe that fungal Chao 1 was the highest values in PTS+CK.

Line 216. How do you know RTS treatments significantly increased …? Did you measure the PERMANOVA? What is the result that indicated the significant increase?

Line 217. The word “Quantities” is not correct. You measured the relative abundance, rather than absolute abundance. How do you quantify the abundance? What is the meaning of quantities?

Line 218. Also here, how do you know significantly decreased? Without statistical analysis, you cannot say significantly… If you did any statistical analysis for the bacterial community compositions, please emphasize here.

Please add information What is the total read of the 16S rRNA sequencing result? How much remaining after filtering? How do you process the sequence read? After the quality control, how many sequences reads remain? You should mention this information of the 16S rRNA sequence at either methods or result parts in your manuscript. It is the main focus of your manuscript. Without this detailed information, it is hard to trust the data and results.

Line 295-296. How do you know RTS treatments increased the bacterial and fungal abundance compared with PTS treatments? The bacterial Chao 1 analysis indicated that PTS+MN has the highest bacteria abundance than RTS. The description didn’t match with the results shown in the figure.

Line 323-324. Which data indicated microbial diversity was significantly influenced by all factors? Which factors were involved in the analysis? Also, the next sentence is not accurate. Not all PTS or RTS treatments were increased/decreased. You also should discuss why PTS+MN showed the highest richness, rather than CK or HN. Is it related to soil nutrition, or other reasons?

---

## Round 0.2 · Major Revisions

The issue of clarity in writing and language edits has been brought up again. Please consider revising it accordingly before resubmission.

Reviewer 3 ·

Basic reporting

Introduction should be presented in present sentence unless examples are cited to support the stated facts and background. Accordingly, authors should rewrite the introduction section from the second paragraph.

In most of the (sub)sections, the language is not clear: the main area of focus for the authors to work on before resubmitting.

Authors should avoid unnecessary usage of comma and phrases that create ambiguity. Also, avoid non-scientific words and contractions like wasn’t, don’t, etc. Grammar should be double checked.

Authors need to clearly state the existing problem(s) related to the land management activities in the area and why the current interventions (treatments) were sought.

Experimental design

In the site description, the last sentence (Line 99-101) is not complete; authors should rewrite it.

Why these levels of N? Authors would justify this.

The experiment was established in 2015 and the samples were collected in 2019? Readers would miss information on whether the treatments were applied only once (in 2015) or repeated every year (season) till 2019. Similarly, whether the soil samples were collected only once (in 2019) or every season after harvest from 2015 to 2019. I think authors need to indicate this in this section.

Validity of the findings

This study addressed important aspects of land management activities on soil properties and microbial communities. Authors have put great effort to acquire rich data and presented the results in tables and figures.

However, the manuscript is not well organized and needs to be rewritten.

Some sentences are not complete or complexed with unnecessary phrases. There is also repetition, and authors would consider combining results and discussions sections to minimize this.

It's better to combine the two subsections: 'Effect of tillage with straw incorporation and N levels on microbial abundance' and 'Effect of tillage with straw incorporation and N levels on microbial composition'. Abundance and composition are both measures of diversity.

Line 205-209: Not clear sentences, please rewrite them.

Line 213: ‘improvements’ is not the right term here. Instead, you can use ‘increment’?

Line 288-290: I couldn’t understand the meaning; the grammar is not correct, either. Consider deleting this sentence.

Line 295: amino acids and amino sugars are not minerals. Authors need to be careful in using terms across the manuscript and avoid using non-scientific words.

---

## Round 0.3 · Minor Revisions

I agree with the reviewer. Please consider revising as suggested.

Reviewer 3 ·

Basic reporting

The manuscript has been improved very well.

Experimental design

Authors responded as to why they chose the N rates in the rebuttal. It would be better if they add this justification in the Experimental Design section.

Validity of the findings

The manuscript is well improved now after the authors have incorporated the comments provided in the first round review.

---

## Round 0.4 · accepted · Accept

Thanks for the revision, a positive recommendation has been made.